# Misperception of the Cardiovascular Risk in Patients with Rheumatoid Arthritis

**DOI:** 10.3390/ijerph17165954

**Published:** 2020-08-17

**Authors:** Jéssica Alonso-Molero, Diana Prieto-Peña, Guadalupe Mendoza, Belén Atienza-Mateo, Alfonso Corrales, Miguel Á. González-Gay, Javier Llorca

**Affiliations:** 1Departamento de Medicina Preventiva y Salud Pública, Universidad de Cantabria—IDIVAL, ES-39008 Santander, Spain; alonsomoleroj@gmail.com; 2CIBER Epidemiology and Public Health (CIBERESP), 28029 Madrid, Spain; 3Epidemiology, Genetics and Atherosclerosis Research Group on Systemic Inflammatory Diseases, Rheumatology Division, Hospital Universitario Marqués de Valdecilla, IDIVAL, ES-39011 Santander, Spain; diana.prieto.pena@gmail.com (D.P.-P.); mateoatienzabelen@gmail.com (B.A.-M.); afcorralesm@hotmail.com (A.C.); 4Instituto Mexicano del Seguro Social, Unidad de Investigación Biomédica 02, Universidad de Colima, 28040 Guadalajara, Mexico; lnmendoza89@gmail.com; 5Cardiovascular Pathophysiology and Genomics Research Unit, School of Physiology, Faculty of Health Sciences, University of the Witwatersrand, Johannesburg 2193, South Africa; 6Facultad de Medicina, Avda., Cardenal Herrera Oria s/n., ES-39008 Santander, Spain

**Keywords:** rheumatoid arthritis, cardiovascular, EULAR-SCORE, Mediterranean diet, physical activity

## Abstract

The risk of cardiovascular (CV) disease and mortality is increased by rheumatoid arthritis (RA). However, data on how RA patients perceive their own CV risk and their adherence to CV prevention factors are scarce. We conducted an observational study on 266 patients with RA to determine whether the perceived CV risk correlates to the objective CV risk, and if it influences their compliance with a Mediterranean diet and physical exercise. The objective CV risk was calculated according to the modified European League Against Rheumatism (EULAR) Systematic Coronary Risk Evaluation (SCORE). The perceived CV risk did not correlate to the objective CV risk. The correlation was even lower when carotid ultrasound was used. Notably, 64.62% of patients miscalculated their CV risk, with 43.08% underestimating it. Classic CV risk factors, carotid ultrasound markers and ESR and CRP showed significant correlation with the objective CV risk. However, only hypertension and RA disease features showed association with the perceived CV risk. Neither the objective CV risk nor the perceived CV risk were associated with the accomplishment of a Mediterranean diet or physical activity. In conclusion, RA patients tend to underestimate their actual CV risk, giving more importance to RA features than to classic CV risk factors. They are not concerned enough about the beneficial effects of physical activity or diet.

## 1. Introduction

Patients with rheumatoid arthritis (RA), a chronic inflammatory disease involving mainly the joints, have an increased risk of cardiovascular (CV) disease and augmented CV mortality when compared with the general population [1,2]. This is the result of a complex compound effect of traditional CV risk factors, chronic inflammation and a genetic component [3,4]. The combination of these factors leads to vascular endothelial damage and accelerated atherosclerosis [5,6]. Several studies have shown that patients with RA have a high prevalence of subclinical atherosclerosis, detected by a number of tools, some of them affordable and reproducible, such as the carotid ultrasound [7,8].

Active treatment of the disease is relevant to decrease the incidence of CV events in patients with RA [9,10]. A patient’s active collaboration in reducing the influence of the classic CV risk factors is also required. However, despite the well-known fact that physical activity or a Mediterranean diet reduce the morbidity and mortality of CV disease and the symptoms of RA [11,12,13], little is known about the adherence to CV prevention factors in patients with RA. In this regard, some data indicate that it is low in the general population [14].

One of the main factors that helps RA patients adopt a healthier lifestyle is the perception of themselves as individuals at high CV risk [15]. However, information related to RA patients’ awareness of their high CV risk compared to people without RA is scarce [16]. Considering that the use of the Systematic Coronary Risk Evaluation (SCORE) model is recommended by the European League Against Rheumatism (EULAR) to classify the CV disease risk of patients with RA into four risk subgroups (low, moderate, high and very high) [17], we may also use this tool to determine how RA patients perceive their own CV risk, how long it is misjudged, or if it influences their lifestyles.

Taking into account these considerations, we assessed a series of patients with RA followed-up at the Rheumatology Outpatient Clinic of a reference center from Northern Spain. We aimed to determine the relationship between the objective and perceived CV risk of these patients as well as establish the patients’ compliance with the Mediterranean diet and physical exercise according to both the objective and perceived CV risk.

## 2. Patients and Methods

We performed a cross-sectional observational study of patients with RA. All patients with RA diagnosis who attended the Rheumatology outpatient clinic for routine follow-up visits between October and December 2018, and agreed to participate, were consecutively included in the study. All of them fulfilled the 2010 ACR/EULAR classification criteria for RA [18]. Patients with cognitive impairment who were unable to complete questionnaires were excluded. The study protocol was approved by the Institutional Review Committee at Hospital Marqués de Valdecilla in Santander, Spain, on 25 May 2012 (Approval Number: 17/2012). All subjects provided informed written consent.

Besides a clinical interview, a carotid ultrasonography (US) examination, which included the measurement of carotid intima-media wall thickness (cIMT) in the common carotid artery and the detection of focal plaques in the extracranial carotid tree, was performed as previously reported [7,19]. Moreover, laboratory testing, including full blood cell count, erythrocyte sedimentation rate (ESR), ultra-sensitivity C- reactive protein (CRP), hepatic and liver functions tests, glucose, insulin and lipid profile, was performed at the time of the assessment.

Medical records of the patients were reviewed to identify the presence of classic CV risk factors (dyslipidemia, diabetes mellitus, hypertension, smoking, overweight or obesity and chronic kidney disease), history of CV events, disease duration, and therapies received by the patients (non-steroidal anti-inflammatory drugs (NSAIDs), glucocorticoids, and conventional or biologic disease modifying anti-rheumatic drugs (DMARDs)) were assessed.

CV risk factors were defined based on current international guidelines. Hypertension was defined as a systolic or a diastolic blood pressure higher than 140 and 90 mmHg, respectively [20]. Dyslipidemia was defined if one of the following factors was present: total cholesterol > 200 mg/dL, triglycerides > 150 mg/dL, HDL-cholesterol < 40 mg/dL in men or <50 mg/dL in women, or LDL-cholesterol > 130 mg/dL [21]. Obesity was defined as a body mass index (BMI) of ≥30 and overweight as a BMI of 25 to <30 [22]. Diabetes mellitus was defined as a fasting plasma glucose ≥ 126 mg/dL, 2-h-postload plasma glucose ≥ 200mg/dL, and HbA1c ≥ 6.5% [23]. Chronic kidney disease was defined as a glomerular filtration rate <60 mL/min/1.73 m^2^ [24].

At that time the CV risk of the patients was calculated according to the SCORE [25] and the modified EULAR SCORE as suggested by the EULAR task force for CV disease prevention in RA modifications [17], and classified as low, moderate, high or very high [17,25]. Also, carotid ultrasound was performed to determine the presence of subclinical atherosclerosis.

Information on the quality of life of the patients was retrieved by GM using the Health Assessment Questionnaire (HAQ) [26,27]. Patients were asked to classify their perception of CV risk as low, moderate, high, or very high in order to make perceived risk comparable to their EULAR SCORE risk [17,25]. All of them were requested to answer this question: “How severe do you think your cardiovascular risk is? Low, moderate, high or very high?”. Patients were asked about their use of a Mediterranean diet using the PREDIMED form (http://www.predimed.es/uploads/8/0/5/1/8051451/p14) [28]. Physical activity was assessed using the Modifiable Activity Questionnaire (http://www.parcph.org/Docs/MAQ.pdf) [29] and was expressed as weekly hours.

### Statistical Analysis

Continuous variables are described as median and interquartile range and categorical variables as number and percentage. The association between perceived CV risk and the updated modified EULAR SCORE [17] was evaluated with Goodman-Kruskal gamma test. Spearman correlation coefficient was estimated to analyze if the CV risk factors and characteristics of RA were associated with objective (modified EULAR SCORE) or perceived CV risk. ANOVA was used to establish the association of Mediterranean diet or weekly hours of physical activity with objective or perceived CV risk. Statistical significance was set in *p* < 0.05.

## 3. Results

Two hundred and sixty-six RA patients were enrolled in this study. Table 1 shows the main demographic and clinical characteristics of these patients. The median age was 60 (53–67) years, 207 (78%) were women and 112 (43%) had a high educational level. The median disease duration from the diagnosis was 10 (5–15.5) years. Ninety-one percent of them had received conventional DMARDs and 46% biologic DMARDs. Fourteen patients (5%) had experienced CV events, 67% of the patients were overweight or obese, 15% were smokers, 43% had hypertension and 66% dyslipidemia.

The median cIMT was 0.667 (0.590–0.773) mm, a 17% presented cIMT > 0.9 mm, and 62% had carotid plaques.

Based on the modified EULAR SCORE, 31.15% of patients were at low risk, 60.38% at moderate risk and 8.46% were at high or very high risk. Interestingly, the perceived CV risk was not associated with the objective CV risk measured by the modified EULAR SCORE. With respect to this, 57.69% self-considered to be at low risk, 28.46% at moderate risk, and 13.85% at high or very high risk (Table 2). Only 35.38% of the subjects perceived correctly the CV risk, whereas 21.54% overestimated and 43.08% underestimated the CV risk.

In a further step, we assessed the effect of including carotid US results in the CV risk stratification of the patients as well as in the perceived CV risk of the patients.

As previously reported [7,19], we observed that 140 (58.8%) patients included in the categories of low and moderate CV risk according to the SCORE were reclassified as having high CV risk after the carotid US assessment (Table 3). Interestingly, the use of carotid US led to a lower correlation between the objective CV risk and the perceived CV risk by the patients.

Each CV risk factor showed a significant correlation with the modified EULAR SCORE. It was also the case for carotid US surrogate markers of CV disease (plaques and cIMT). However, only hypertension was associated with perceived CV risk (Table 4).

When association between RA characteristics and the modified EULAR SCORE or CV risk perception were assessed, only ESR and CRP showed a significant correlation with the modified EULAR SCORE. However, the visual analogical scale (VAS) and HAQ as well as the presence of positive anti-CCP antibodies showed a significant association with the perceived CV risk (Table 5).

Neither the modified EULAR score nor the perceived CV risk were associated with current use of the Mediterranean diet or the physical activity performance (Table 6).

## 4. Discussion

The results from the present study indicate that patients with RA do not perceive the increased risk of CV disease associated with this condition. In keeping with our findings, it was suggested that patients with RA do not seem to be aware of this problem [14] and often have trouble perceiving their actual CV risk [15,16]. With respect to this, a recent systematic review [16] identified only six studies that assessed the degree of the perceived risk of CV disease development in patients with RA, comparing it with EULAR SCORE or with classic CV risk factors. Overall, the number of patients included in these studies was close to 480 subjects with RA. This review concluded that patients with RA tend to misjudge their risk of developing CV disease, usually underestimating it [16].

Our results support these findings. Patients with RA are not aware of their actual CV risk. Their perception of CV risk does not correlate to their objective CV risk measured by the updated modified EULAR SCORE and the presence of classic CV risk factors. In our study, two thirds of RA patients misjudged their risk of CV disease, and almost half underestimated it. This supports the idea by Fronlund et al. [30] and Van Breukelen-Van der Stoep et al. [31], who reported that most patients with RA have limited knowledge about CV disease and underestimate their risk. However, these results are not entirely in agreement with those by Boo et al. [15], who reported that the percentage of patients who overestimated CV risk was approximately twice that of patients who underestimated it.

RA patients are also not sufficiently aware of the relevance of physical activity or healthy dietary habits to prevent CV events. Boo et al. [15,26] observed that there is a high percentage of sedentary population independently of their objective CV risk. They stated that patients did not consider physical inactivity and obesity as CV risk factors [15]. For this reason, even when patients know that they are at high risk of developing CV disease, their adherence to physical activity or diet is less than to other beneficial interventions such as antihypertensive drugs and/or lipid-lowering drugs [31]. Our results confirmed that neither the objective CV risk nor the perceived CV influences the accomplishment of the Mediterranean diet or daily physical activity.

There were a number of limitations in our study. The main one was its cross-sectional nature. Another potential limitation was that the perception of the CV risk of the patients is subjective and it could yield to great variation in terms of severity from patient to patient. In addition, we did not stratify patients according to DMARD intake or biologic therapy and glucocorticoid use. However, our article makes some new contributions. To the best of our knowledge, this is the first study assessing the correlation between classic CV risk factors and the perceived CV risk. In this regard, all classic CV risk factors were associated with the objective EULAR SCORE CV risk, but not with the perceived CV risk. In contrast, we observed that the perceived CV risk was associated with some RA disease features, such as anti-CCP, VAS or HAQ. Interestingly, the objective CV risk was only associated with ESR and CRP levels. This supports the results of our previous studies that showed that the increased CV mortality in patients with RA from Northwest Spain was associated with chronic inflammation [32]. The fact that the perceived CV risk was not associated with classic CV risk factors is challenging. It might suggest that patients with RA are aware of their own CV risk based on certain features of the disease, such as pain or disability, rather than on classical CV risk factors.

In conclusion, our results suggest that patients with RA tend to underestimate the risk of developing CV disease. In addition, they are not sufficiently concerned about the beneficial effects of physical activity and the Mediterranean diet, giving more importance to RA disease features than to classic CV risk factors.

## 5. Conclusions

In conclusion, our results suggest that patients with RA tend to underestimate the risk of developing CV disease. In addition, they are not sufficiently concerned about the beneficial effects of physical activity and the Mediterranean diet, giving more importance to RA disease features than to classic CV risk factors.

## Figures and Tables

**Table 1 ijerph-17-05954-t001:** Description of 266 patients with rheumatoid arthritis.

	Category	Number (%) or Median (IQR)
Demographic Characteristic		
Age		60 (53–67)
Sex	Women	207 (78%)
	Men	59 (22%)
Educational level	Less than Primary	4 (1.5%)
	Primary	91 (34%)
	Secondary	84 (32%)
	University	28 (11%)
	Unknown	59 (22%)
Current working activity	No	118 (44%)
	Yes	90 (34%)
	Unknown	58 (22%)
Rheumatoid arthritis characteristics		
Years from RA diagnosis		10 (5–15.5)
NSJ28		1 (0–3)
NTJ28		0 (0–3)
Global patient evaluation		30 (20–60)
DAS28 (ESR)		2.6 (1.7–3.7)
DAS28 (CRP)		2.6 (1.8–3.6)
HAQ		0.75 (0.13–1.25)
Conventional DMARDs *		243 (91%)
Biologic DMARDs *		123 (46%)
NSAIDs *		202 (76%)
ESR		9 (3–18)
CRP		2 (0.5–5)
Cardiovascular risk		
History of cardiovascular events		14 (5%)
BMI	Overweight	100 (38%)
	Obesity	76 (29%)
Chronic kidney disease		13 (5%)
Diabetes mellitus		24 (9%)
Hypertriglyceridemia		53 (20%)
Hypertension		115 (43%)
Dyslipidemia		175 (66%)
Current smoking		40 (15%)
SCORE		1 (0–2)
SCORE–EULAR		1 (0–2)
SCORE–EULAR 2016		1.5 (0–3)
cIMT		0.667 (0.590–0.773)
Carotid plaque		160 (60%)
PVW		6.9 (6.1–8.4)
Carotid arterial stiffness		11 (4%)

IQR: interquartile range; NSJ: number of swollen joints; NTJ: number of tender joints; DAS28: Disease Activity Score 28; ESR: erythrocyte sedimentation rate; CRP: C-reactive Protein. HAQ: Health Assessment Questionnaire; DMARDs: disease modifying anti-rheumatic drugs; NSAIDs: non-steroidal anti-inflammatory drugs; ESR: erythrocyte sedimentation rate; BMI: Body mass index; SCORE: Systematic Coronary Risk Evaluation; EULAR: European League Against Rheumatism; cIMT: carotid intima-media thickness; PVW: pulse wave velocity. * At any time from disease diagnosis to the time of assessment.

**Table 2 ijerph-17-05954-t002:** Relationship between perceived cardiovascular risk and European League Against Rheumatism (EULAR) Systematic Coronary Risk Evaluation (SCORE)-estimated cardiovascular risk.

	Perceived Cardiovascular Risk			
EULAR SCORE 2016	Low	Moderate	High	Total
**Low (0%)**	47 (58.02%)	21 (25.93%)	13 (16.05%)	81 (100%)
**Moderate (1–4.9%)**	91 (57.96%)	44 (28.03%)	22 (14.01%)	157 (100%)
**High (≥5%)**	12 (54.54%)	9 (40.91%)	1 (4.54%)	22 (100%)
**Total**	150 (57.69%)	74 (28.46%)	36 (13.85%)	260 (100%)

Goodman-Kruskal gamma = −0.0169, Asymptotic Standard Error = 0.103, *p* = 0.57.

**Table 3 ijerph-17-05954-t003:** Relationship between original EULAR SCORE 2016 and SCORE reclassified by the carotid US assessment.

EULAR SCORE 2016	SCORE Reclassified			
	Low	Moderate	High	Total
**Low**	54 (100%)	0 (0%)	27 (16.67%)	81 (31.15%)
**Moderate**	0 (0%)	44 (100%)	113 (69.75%)	157 (60.38%)
**High**	0 (0%)	0 (0%)	22 (13.58%)	22 (8.46%)
**Total**	54 (100%)	44 (100%)	162 (100%)	260 (100%)

**Table 4 ijerph-17-05954-t004:** Spearman correlation between cardiovascular risk factors and EULAR SCORE 2016 and cardiovascular risk perception.

CV Risk Factor	EULAR SCORE 2016	Perceived CV Risk
**Obesity**	0.1885 (0.003)	0.1179 (0.07)
**CKD**	0.1407 (0.03)	0.0375 (0.56)
**Diabetes mellitus**	0.1487 (0.02)	0.0796 (0.22)
**Hypertriglyceridemia**	0.1674 (0.009)	0.0399 (0.54)
**Dyslipidemia**	0.1654 (0.01)	0.0926 (0.15)
**Hypertension**	0.3778 (<0.001)	0.1362 (0.03)
**Carotid plaques**	0.4453 (<0.001)	0.0883 (0.17)
**BMI**	0.3106 (<0.001)	0.1157 (0.07)
**cIMT**	0.4746 (<0.001)	−0.0084 (0.90)

BMI: body mass index, cIMT: carotid intima-media thickness, CKD: Chronic kidney disease, CV: cardiovascular.

**Table 5 ijerph-17-05954-t005:** Spearman correlation between rheumatoid arthritis characteristics and SCORE-EULAR 2016 and cardiovascular risk perception.

Rheumatoid Arthritis Characteristics	Modified EULAR SCORE	Perceived Cardiovascular Risk
**Rheumatoid factor**	0.0347 (0.60)	0.0990 (0.14)
**Anti-CCP**	−0.0593 (0.37)	0.1358 (0.04)
**NTJ28**	−0.0614 (0.36)	0.0337 (0.61)
**NSJ28**	−0.0411 (0.54)	0.0525 (0.43)
**VAS (0–100)**	−0.0890 (0.18)	0.1627 (0.01)
**DAS28-ESR**	−0.0038 (0.95)	0.0900 (0.18)
**DAS28-CRP**	−0.0445 (0.50)	0.0725 (0.28)
**HAQ**	−0.0550 (0.41)	0.1593 (0.02)
**ESR**	0.1499 (0.02)	0.0865 (0.19)
**CRP**	0.1647 (0.01)	0.0421 (0.53)

CCP: citrullinated peptides, CRP: ultra-sensitivity C- reactive protein, DAS28: Disease Activity Score in 28 joints, ESR: erythrocyte sedimentation rate, HAQ: Health Assessment Questionnaire, NST: number of swollen joints, NTJ: number of tender joints, VAS: visual analogic scale.

**Table 6 ijerph-17-05954-t006:** Accomplishment of Mediterranean diet and physical exercise according to perceived and SCORE-estimated cardiovascular risks.

EULAR SCORE 2016	PREDIMED Score	*P*	Weekly Hours of Physical Activity	*P*
0	8.1 ± 2.1	0.92	3.4 ± 3.3	0.19
1.5	8.3 ± 1.6		3.8 ± 3.4	
3	8.2 ± 2.3		3.1 ± 2.8	
4.5	8.1 ± 1.7		5.2 ± 3.7	
≥6	8.4 ± 2.2		4.2 ± 3.6	
Perceived cardiovascular risk				
Low	8.3 ± 1.8	0.43	3.9 ± 3.3	0.57
Moderate	8.2 ± 2.0		3.4 ± 3.4	
High	7.7 ± 2.2		3.2 ± 3.6	
Very high	8.0 ± 2.0		3.1 ± 3.4

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
