# Peer review of "Misperception of the Cardiovascular Risk in Patients with Rheumatoid Arthritis"

_ijerph, 2020, doi:10.3390/ijerph17165954_

Round 1

Reviewer 1 Report

I read with interest the study by Alonso-Molero et al reporting the perception of CV risk by RA patients. In this paper, 266 RA patients were asked to rate their own CV risk as low, moderate, high or very-high. This self-rating was then confronted to the objective CV risk calculated by means of the SCORE and the revised version by EULAR. The Authors found that RA patients commonly fail to estimate their CV risk.

Overall, the study is well written and the results are interesting although not completely novel.

I have a few comments for the Authors:

Major points

  1. The Methods section needs improvement. i) Was the study approved by an IRB? Was informed consent obtained? ii) Is this a cross-sectional study? How were the patients selected? How were patients included or excluded? iii) How were CV risk factor defined? It may be useful to provide references. iv) Please report the precise question used to ask the perceived risk to the patients. v) How was the frequency of physical activity scored and summarized?
  2. Statistical analysis: i) Please add a paragraph on how categorical and numerical variables were summarized. ii) Statistical significance threshold must be clarified. iii) Correlation. Why did the Authors use Spearman correlation with categorical variables such as CV risk factors? If they are interpreted dichotomously I would rather use Point-biserial correlation analysis or other measures of association (i.e. Chi square test using risk categories instead of numeric scores). 
  3. The Authors should discuss some important limitations of the study in the Discussion section.

Minor points

Table 1. CRP is misspelled (PCR). With regard to therapies, are they current or past therapies? Please clarify.

Add abbreviations under the Tables.

English needs some minor grammar checks.

Reviewer 2 Report

The topic is interesting and worth further investigation. 

Some methodological aspects require some clarifications:

1) inclusion criteria, study design (retrospective, chart-driven? prospective?) should be better detailed. 

2) the methods section is somehow confusing. The Authors should clearly state which tools have been used for actual CV risk assessment and which for estimation of perceived CV risk, eventually creating different subheadings.

In detail, the SCORE is a simple, office-based screening tool that predicts the probability of developing CVD in the next 10 years based on some clinical and demographic characteristics. The authors should detail how this tool was applied in estimating “perceived CV risk”. 

2) Polypharmacotherapy is another important factor when assessing perceived health status variables. Has this aspect evaluated? Have patients been stratified according to the number of medications they assume daily? (e.g. those taking more mediation are more aware of their health status?)

Round 2

Reviewer 1 Report

The Authors have addressed the criticism. I have no further comments.